# Protein Structure Representation Learning through Orientation-Aware Graph Neural Networks

## Abstract

By folding to particular 3D structures, proteins play a key role in living beings. To learn meaningful representation from a protein structure for downstream tasks, not only the global backbone topology but the local fine-grained orientational relations between amino acids should also be considered. In this work, we propose the Orientation-Aware Graph Neural Networks (OAGNNs) to better sense the geometric characteristics in protein structure (e.g. inner-residue torsion angles, inter-residue orientations). Extending a single weight from a scalar to a 3D vector, we construct a rich set of geometric-meaningful operations to process both the classical and $SO(3)$ representations of a given structure. To plug our designed perceptron unit into existing Graph Neural Networks, we further introduce an equivariant message passing paradigm, showing superior versatility in maintaining $SO(3)$-equivariance at the global scale. Experiments have shown that our OAGNNs have a remarkable ability to sense geometric orientational features compared to classical networks. OAGNNs have also achieved state-of-the-art performance on various computational biology applications related to protein 3D structures.

## 1 Introduction

Built from a sequence of amino-acid residues, a protein performs its biological functions by folding to a particular conformation in 3D space. Therefore, untilizing such 3D structures accurately is the key for downstream analysis. While we have witnessed remarkable progress in protein structure predictions (Rohl et al., 2004; Källberg et al., 2012; Baek et al., 2021; Jumper et al., 2021), another thread of tasks with protein 3D structures as input starts to draw a great interest, such as function prediction (Hermosilla et al., 2020; Gligorijević et al., 2021), decoy ranking (Lundström et al., 2001; Kwon et al., 2021; Wang et al., 2021), protein docking (Duhovny et al., 2002; Shulman-Peleg et al., 2004; Gainza et al., 2020; Sverrisson et al., 2021), and driver mutation identification (Lefèvre et al., 1997; Antikainen & Martin, 2005; Li et al., 2020; Jankauskaitė et al., 2019).

Most existing works in modeling protein structures directly borrow models designed for other applications, including 3D-CNNs (Ji et al., 2012) in computer vision, Transformers (Vaswani et al., 2017)) from natural language processing, and GNNs (Kipf & Welling, 2016) in data mining. Though compatible with general objects, these models have overlooked the subtleties in the fine-grained geometries, which are much more essential in protein structures. For instance, given an amino acid in the protein structure, as shown in Figure 1, the locations of four backbone atoms (carbon, nitrogen, and oxygen) determine a local skeleton, and different residues interact with each other through performing specific orientations between their local frames, either of which have important impacts on the protein structure and its function (Nelson et al., 2008).

Recent attempts in building geometric-aware neural networks mainly focus on baking *3D rigid transformations* into network operations, leading to the area of $SO(3)$-invariant and equivariant networks. One representative work is the Vector Neuron Network (VNN) (Deng et al., 2021), which achieves $SO(3)$-equivariance on point clouds by generalizing scalar neurons to 3D vectors. Another work is the GVP-GNN (Jing et al., 2021) that similarly vectorizes hidden neurons in GNN and demonstrates better prediction accuracy on protein design and quality evaluation tasks. However,

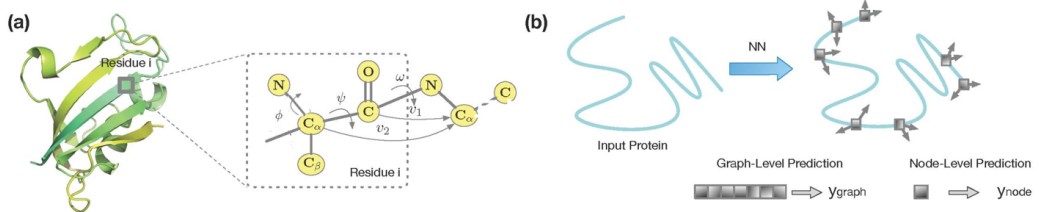

Figure 1: Overview (a) Each amino acid has its own **rigid backbone** with four heavy atoms, and can be represented by both lists of scalar and vector features. (b) Tasks associated with the protein 3D structure. **Graph-level** tasks consider the whole protein structures, and **Node-level** tasks operate on specific residues.

these two models can only adopt *linear combinations* of input vectors, which significantly limits their modeling capability. A simple example is that, given two input vector features $v_1$ and $v_2$, the outputs $w_1 v_1 + w_2 v_2$ through one linear layer is constrained in the 2D plane spanned by $v_1, v_2$ even after applying their scalar-product non-linearities. That is, VNN-based models are limited in perceiving orientational features, which have been proven crucial for proteins to perform their functions and interact with other partners (*e.g.* inner-residue torsion angles, inter-residue orientations) (Nelson et al., 2008; Voet & Voet, 2010; Xu & Berger, 2006; Alford et al., 2017).

To achieve more sensitive geometric orientation awareness, we propose a *Directed Weight ($\vec{\mathbf{W}}$) perceptrons* by extending not only the hidden neurons but also the weights from scalars to 3D vectors, naturally saturating the entire network with 3D structure information in the Euclidean space. Directed weights support a set of geometric-meaningful operations on both the vector neurons (vector-list features) and the classical (scalar-list) latent features, and perform flexible non-linear integration of the hybrid scalar-vector features. As protein structures are naturally attributed proximity graphs, we introduce a new *Equivariant Message Passing Paradigm* on protein graphs, to connect the $\vec{\mathbf{W}}$-perceptrons with the graph learning models by using rigid backbone transformations for each amino acid, which provides a versatile framework for bringing the biological suitability and flexibility of the GNN architecture.

To summarize, our key contributions include:

- We propose a new network unit based on the *Directed Weights* for capturing fine-grained geometric relations, especially for the subtle orientational details in proteins.
- We construct an *Equivariant Message Passing* paradigm based on protein graphs.
- Our overall framework, the *Orientation-Aware Graph Neural Networks*, is versatile in terms of compatibility with existing deep graph learning models, making them biologically suitable with minimal modifications to existing GNN models.

## 2 RELATED WORK

**Representation learning on protein 3D structure.** Early approaches rely on hand-crafted features extracted and statistical methods to predict function annotations (Schaap et al., 2001; Zhang & Zhang, 2010). Deep learning has been found to achieve success then. 3D CNNs are first proposed to process protein 3D structures by scanning atom-level features relying on multiple 3D voxels. One of the representative works (Derevyanko et al., 2018) adopts a 3D CNN-based model for assessing the quality of the predicted structures. 3D CNNs also shed light on other tasks such as interface prediction (Townshend et al., 2019; Amidi et al., 2018). People also extend them to spherical convolutions (Gainza et al., 2020; Sverrisson et al., 2021; Hermosilla Casajus et al., 2021), to the Fourier space (Zhemchuzhnikov et al., 2022) and the 3D Voronoi Tessellation space (Igashov et al., 2021). Graph Convolutional Networks (Kipf & Welling, 2016) have also been adopted to capture geometric and biochemical interactions between residues (Ying et al., 2018; Gao & Ji, 2019; Fout, 2017), and have been shown to achieve great performance on function prediction (Li et al., 2021), protein design (Strokach et al., 2020) and binding prediction (Vecchio et al., 2021). Recently, transformer-based

methods (Vaswani et al., 2017) have a trend to replace conventional methods in other bioinformatics tasks (Ingraham et al., 2019; Baek et al., 2021; Jumper et al., 2021; Cao et al., 2021).

**Equivariant neural networks.** Equivariance is an important property, for generalizing to unseen conditions in geometric learning. (Cohen & Welling, 2016; Weiler et al., 2018; Köhler et al., 2020; Satorras et al., 2021; Thomas et al., 2018; Fuchs et al., 2020; Anderson et al., 2019; Gasteiger et al., 2019b; Batzner et al., 2021; Eismann et al., 2021). Methods such as Tensor Field Network (Thomas et al., 2018) and Cormorant (Anderson et al., 2019), have been developed to generate irreducible representations for achieving rotation equivariance in 3D. In comparison to Tensor Filed Network with complex tensor products, the Vector Neuron Network (VNN) achieves rotation equivariance in a much simpler way, which generalizes the values of hidden neurons from scalars to 3D vectors (Deng et al., 2021). GVP-GNN has also been introduced for learning protein representations by featuring geometric factors as vectors (Jing et al., 2020). Message passing in GNNs for vector representations using equivariant features have also been explored (Schütt et al., 2021; Luo et al., 2022). However, to guarantee rotation equivariance, they can only linearly combine 3D vectors, in essence, limiting their geometric representing capacity. Another line of methods condition filters on invariant scalar features for maitaining equivariance (Schütt et al., 2017; Gasteiger et al., 2019a; Liu et al., 2021).

## 3 DIRECTED WEIGHT PERCEPTRON

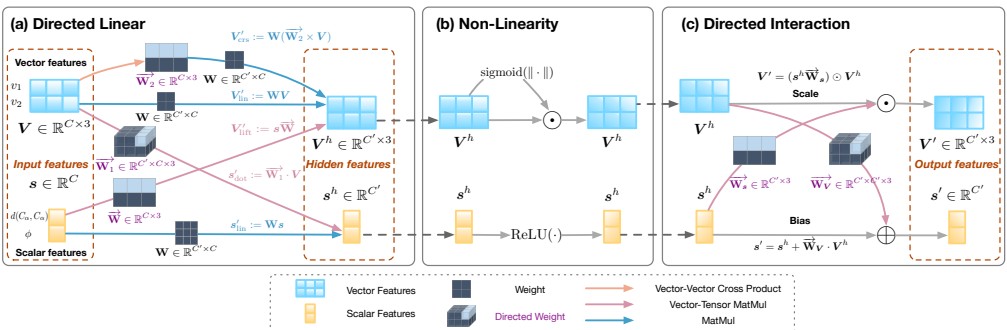

Figure 2: Model Details. A 1-layer DWP consists of three following modules. Both the input, hidden and output are tuples. (a) **Directed Linear Module** applies multiple geometric operations to update scalar and vector features in four different ways with normal and directed weights. (b) **Non-Linearity Module** employs ReLU and sigmoid functions for scalar and vector features. (c) **Directed Interaction Module** updates the features by using one another as updating parameters after non-linearity in another way.

A protein is modeled as a KNN-graph $\mathcal{G} = (\mathcal{V}, \mathcal{E})$ where each node $u \in \mathcal{V}$ corresponds to one amino acid, characterized by a scalar-vector tuple $h_u = (s_u, V_u)$ with $s_u \in \mathbb{R}^C, V_u \in \mathbb{R}^{C \times 3}$, and the edges are constructed spatially by querying its $k$-nearest neighbours in the space. The edge features $\mathcal{E} = \{e_{ij}\}_{i \neq j}$, which represent edge connected node $i$ and $j$, are also multi-channel scalar-vector tuples. For simplicity, we set the channel numbers for scalar and vector features as the same, but they can be different. In practice, scalar, vector features are constructed based on a given protein structure and are kept separate. To make them compatible, the channel numbers of scalars are larger than the vectors. Details can be found in the Appendix Section B.1.

### 3.1 DIRECTED WEIGHTS

Classical neural networks only consider scalar-list features of the form $s \in \mathbb{R}^C$, with each layer transforming them with a weight matrix $\mathbf{W} \in \mathbb{R}^{C' \times C}$ and a bias term $\boldsymbol{b} \in \mathbb{R}^{C'}$:

$$s' = \mathbf{W}s + \mathbf{b} \tag{1}$$

Although has been proved to be a universal approximator, such a layer has intrinsically no geometric interpretation in the 3D shape space, and even the simplest 3D rigid transformation on the input can result in unpredictable changes of network outputs. Recent attempts have lifted neuron features from

scalar-lists to vector-lists $V \in \mathbb{R}^{C \times 3}$ with a linear operation mapping to $V' \in \mathbb{R}^{C' \times 3}$ (Deng et al., 2021; Jing et al., 2020):

$$V' = \mathbf{W}V \tag{2}$$

Under this construction, SO(3)-actions in the latent space are simply matrix multiplications, which establishes a clear 3D Euclidean structure. However, to maintain input-output consistency under rigid transformations (e.g. rotation and translation), their operations are still limited to classical linear combinations weighted by $\mathbf{W}$. To define *operations* that are more adaptive to the geometrically meaningful features, we introduce the *Directed Weights*, that we can define any tensor with the last dimension of 3 as a directed weight matrix, for example, $\vec{\mathbf{W}} \in \mathbb{R}^{C' \times C \times 3}$, seen as a stack of geometric meaningful vectors lying in the 3D Euclidean Space, and can be acted by SE(3) group from the right.

## 3.2 $\vec{\mathbf{W}}$-OPERATORS

With weights and features both equipped with 3D vector representations, we can design a set of *geometric operators* $\square$ (e.g. $\cdot$ or $\times$) with learnable $\vec{\mathbf{W}}$ as parameters in neural networks, which can operate on both scalar-list features $s \in \mathbb{R}^C$ and vector-list features $V \in \mathbb{R}^{C \times 3}$

$$\vec{\mathbf{W}}\square s, \quad \vec{\mathbf{W}}\square V \tag{3}$$

**Geometric vector operations.** Let $\mathbf{W} \in \mathbb{R}^{C' \times C}$ be a conventional scalar weight matrix, $\vec{\mathbf{W}}_1 \in \mathbb{R}^{C' \times C \times 3}$ and $\vec{\mathbf{W}}_2 \in \mathbb{R}^{C \times 3}$ be directed weight tensors. Beyond linear combinations, We can define two operations that leverage geometric information:

$$s'_{\text{dot}}(V; \vec{\mathbf{W}}_1) = \vec{\mathbf{W}}_1 \cdot V \qquad \in \mathbb{R}^{C'} \tag{4}$$

$$V'_{\text{crs}}(V; \mathbf{W}, \vec{\mathbf{W}}_2) = \mathbf{W}(\vec{\mathbf{W}}_2 \times V) \quad \in \mathbb{R}^{C' \times 3} \tag{5}$$

Here $s'_{\text{dot}}$ transforms $C$ vector features to $C'$ scalars using dot-product with directed weights, detailedly, $s'_i = \sum_{j,k} \vec{\mathbf{W}}_1^{i,j,k} V^{j,k}$, which explicitly measures angles between vectors, and this operator brings network the ability to accurately sense the orientational features. In $V'_{\text{crs}}$, a vector crosses a directed weight before being projected onto the hidden space. While the output of plain linear combinations between two vectors $v_1$ and $v_2$ can only lie in the plane $w_1 v_1 + w_2 v_2$, cross-product in Equation 5 creates a new direction outside the plane, which is crucial in 3D modelling. For instance, the side-chain angles of a given residue could be largely determined by its $C_\alpha$ and $C_\beta$ vectors, but may lie on the direction perpendicular to the space constructed by the two vectors.

**Scalar lifting.** In addition, a directed weight $\vec{\mathbf{W}} \in \mathbb{R}^{C \times 3}$ can lift scalars to vectors by adopting the following operation, the $ij$ th entry of $s\vec{W}$ is $s_i \vec{W}_{ij}$

$$V'_{\text{lift}}(s; \vec{\mathbf{W}}) = s\vec{\mathbf{W}} \quad \in \mathbb{R}^{C \times 3} \tag{6}$$

This maps each scalar to a particular vector, enabling inverse transformations or information bottlenecks from $\mathbb{R}$ to $\mathbb{R}^3$. An intuition example is that, we can map a scalar representing the distance between two amino acids to a vector pointing from one amino acid to the other, leading to more biological meaningful representations for the protein fragment.

**Linear combinations.** In the end, we keep the linear combination operations with scalar weights for both scalar and vector features:

$$s'_{\text{lin}}(s; \mathbf{W}) = \mathbf{W}s \quad \in \mathbb{R}^{C'} \tag{7}$$

$$V'_{\text{lin}}(V; \mathbf{W}) = \mathbf{W}V \quad \in \mathbb{R}^{C' \times 3} \tag{8}$$

While these operators enables a more flexible network design, equivariance to rigid transformations is broken if considering more complex functions beyond linear combinations. We will introduce a globally equivariant paradigm to tackle this issue in Section 4.

## 3.3 $\vec{\mathbf{W}}$-PERCEPTRON UNIT

Now we combine all the $\vec{\mathbf{W}}$-operators together, assembling them into what we call Directed Weight Perceptrons ($\vec{\mathbf{W}}$-perceptrons). A $\vec{\mathbf{W}}$-perceptron unit is a function mapping from $u = (s, V)$ to

another scalar-vector tuple $u' = (s', V')$, which can be stacked as network layers to form multi-layer perceptrons.

A single unit comprises three modules in a sequential way: the **Directed Linear** module, the **Non-Linearity** module, and the **Directed Interaction** module (Figure 2).

**Directed linear module.** The output hidden representations derived from the set of operations introduced previously are concatenated and projected into another hidden space of scalars and vectors separately:

$$s^h = \mathbf{W}_s [s'_{\text{dot}}, s'_{\text{lin}}] \qquad \in \mathbb{R}^{C'} \qquad (9)$$

$$V^h = \mathbf{W}_V [V'_{\text{crs}}, V'_{\text{lift}}, V'_{\text{lin}}] \quad \in \mathbb{R}^{C' \times 3} \qquad (10)$$

Here $\mathbf{W}_s \in \mathbb{R}^{C' \times (C'+C')}$ and $\mathbf{W}_V \in \mathbb{R}^{C' \times (C'+C+C')}$ are scalar weight matrices. In other word, the five separate updating functions allow transformation from scalar to scalar (Equation 7), scalar to vector (Equation 6), vector to scalar (Equation 4), and vector to vector (Equation 8, Equation 5), boosting the model's capability to reason in 3D space.

**Non-linearity module.** We then apply the non-linearity module to the hidden representation $(s^h, V^h)$. Specifically, we apply standard ReLU non-linearity (Nair & Hinton, 2010) to the scalar components. For the vector representations, following (Weiler et al., 2018; Jing et al., 2020; Schütt et al., 2021), we compute a sigmoid activation on the L2-norm of each vector and multiply it back to the vector entries accordingly:

$$s^h \leftarrow \text{ReLU}(s^h), \quad v^h_{ij} \leftarrow v^h_{ij} \cdot \text{sigmoid}(\|v^h_i\|_2) \qquad (11)$$

where $v^h_i \in \mathbb{R}^3$ are the vector columns in $V^h$ and $v^h_{ij}$ are their entries.

**Directed interaction module.** Finally we introduce the Directed Interaction module, integrating the hidden features $s^h$ and $V^h$ into the output tuple $(s', V')$

$$s' = s^h + \vec{\mathbf{W}}_V \cdot V^h \quad \in \mathbb{R}^{C'} \qquad (12)$$

$$V' = (s^h \vec{\mathbf{W}}_s) \odot V^h \quad \in \mathbb{R}^{C' \times 3} \qquad (13)$$

Here $\vec{\mathbf{W}}_V, \vec{\mathbf{W}}_s$ are directed weight matrices with sizes $C' \times C' \times 3$ and $C' \times 3$, respectively, $\odot$ denotes element-wise multiplication for two matrices. This module establishes a connection between the scalar and vector feature components, facilitating feature blending. Specifically, Equation 12 dynamically determines how much the output should rely on scalar and vector representations, and Equation 13 weights a list of vectors using the scalar features as attention scores.

## 4 ORIENTATION-AWARE GRAPH NEURAL NETWORKS

To achieve SO(3)-equivariance without the loss of modeling capacity, we introduce an *equivariant message passing paradigm* on protein graphs, to easily plug our versatile $\vec{\mathbf{W}}$-Perceptron into any existing graph learning framework, which makes network architectures free from equivariant constraints (Section 4.1). The integrated models, called *Orientation-Aware Graph Neural Networks*, can not only accurately model the essential orientational features but also maintain the rotation equivariance efficiently. We also design multiple variants in analogy to other graph neural networks Section 4.2.

### 4.1 EQUIVARIANT MESSAGE PASSING ON PROTEINS

A function $f : \mathbb{R}^3 \rightarrow \mathbb{R}^3$ is SO(3)-equivariant (rotation-equivariant) if any rotation matrix $R \in \mathbb{R}^{3 \times 3}$ applied to the input vector $x$ leads to the same transformation on the output $f(x)$:

$$R f(x) = f(Rx) \qquad (14)$$

Such equivariant property can be achieved on a protein graph with local orientations, which is naturally defined from its biological structure (Ingraham et al., 2019). Specifically, each amino acid

node $u_i \in \mathcal{V}$ has four backbone heavy atom $(C_\alpha^u, C^u, N^u, O^u)$, defining a local frame $\boldsymbol{O}_u$ as:

$$\boldsymbol{x}_u = N^u - C_\alpha^u \in \mathbb{R}^3, \quad \boldsymbol{y}_u = C^u - C_\alpha^u \in \mathbb{R}^3 \tag{15}$$

$$\boldsymbol{O}_u = \left[ \frac{\boldsymbol{x}_u}{\|\boldsymbol{x}_u\|_2}, \frac{\boldsymbol{y}_u}{\|\boldsymbol{y}_u\|_2}, \frac{\boldsymbol{x}_u}{\|\boldsymbol{x}_u\|_2} \times \frac{\boldsymbol{y}_u}{\|\boldsymbol{y}_u\|_2} \right]^\top \in \mathbb{R}^{3\times3} \tag{16}$$

The local frame $\boldsymbol{O}_u$ is a rotation matrix that maps a 3D vector from the local to the global coordinate system. An equivariant message passing paradigm then emerges through transforming node features back and forth between adjacent local frames. Formally, give an amino acid $u$ with hidden representation $h = (\boldsymbol{s}, \boldsymbol{V})$, let $f_l$ and $f_g$ be transformations on the vector feature $\boldsymbol{V}$ from and to the global coordinate system:

$$f_l(h, \boldsymbol{O}_u) := (\boldsymbol{s}, \boldsymbol{V}\boldsymbol{O}_u^\top), \quad f_g(h, \boldsymbol{O}_u) := (\boldsymbol{s}, \boldsymbol{V}\boldsymbol{O}_u) \tag{17}$$

The message passing update for node $u$ from layer $l$ to layer $l+1$ is performed in the following steps, the $[\quad]$ notations indicates the concatenation of different hidden representations along the channel-wise dimension:

1. Transform the neighbourhood vector representations of $u$ into its local coordinate system $\boldsymbol{O}_u$.

2. For each adjacent node $v$, compute the message $m_u^{l+1}$ on edge $(u, v) \in \mathcal{E}$ with an multi-layer $\vec{\mathbf{W}}$-perceptron $\mathcal{F}$ to the aggregated node and edge feature $[h_u^l, h_v^l, e_{uv}^l]$ (Equation 18).

3. Update node feature $h_u^l$ with another multi-layer $\vec{\mathbf{W}}$-perceptron $\mathcal{H}$, and transform it back to the global coordinate system (Equation 19).

This paradigm achieves SO(3)-equivariance in a very general sense with no constraints on $\mathcal{F}, \mathcal{H}$. The formal proof of equivariance is presented in the Appendix Section A.

$$m_u^{l+1} = \sum_{v \in \mathcal{N}_u} \mathcal{F}(f_l([h_u^l, h_v^l, e_{uv}^l], \boldsymbol{O}_u)) \tag{18}$$

$$h_u^{l+1} = f_g(\mathcal{H}(h_u^l, m_u^{l+1}), \boldsymbol{O}_u) \tag{19}$$

## 4.2 Variants of Orientation-Aware Graph Neural Networks

By integrating the $\vec{\mathbf{W}}$-Perceptrons with equivariant message passing, we propose multiple variants of entire *Orientation-Aware Graph Neural Networks* to boost the design space for different tasks.

**OA-GCN.** This is the one described and implemented in the previous subsection with Equation 18 and Equation 19.

**OA-GIN.** We adopt graph isomorphism operator Xu et al. (2018) with learnable weighing parameter $\varepsilon$ for tunable skip connections,

$$m_u^{l+1} = (1 + \varepsilon)f_l(h_u^l, \boldsymbol{O}_u) + \sum_{v \in \mathcal{N}_u} \mathcal{F}(f_l(h_v^l, \boldsymbol{O}_u)) \tag{20}$$

**OA-GAT.** In comparison to GCN and GIN, it is not trivial to incorporate with the Graph Attention Network (GAT) (Veličković et al., 2018), as residues interact with each other unevenly in proteins. In particular, we define separate attentions for the scalar and vector representations

$$(\boldsymbol{s}_u^l, \boldsymbol{V}_u^l) = f_l(h_u^l, \boldsymbol{O}_u) \tag{21}$$

$$\boldsymbol{s}_u' = \sum_{v \in \mathcal{N}_u \cup \{u\}} \alpha_v^s \boldsymbol{s}_v^l, \quad \boldsymbol{V}_u' = \sum_{v \in \mathcal{N}_u \cup \{u\}} \alpha_v^v \boldsymbol{V}_v^l \tag{22}$$

$$h_u^{l+1} = f_g(\mathcal{H}((\boldsymbol{s}', \boldsymbol{V}')), \boldsymbol{O}_u) \tag{23}$$

The attention scores $\alpha^s, \alpha^v$ are the softmax values over the inner products of all neighboring source-target pairs defined as the follows:

$$\alpha_v^s = \frac{\exp\left(\langle \boldsymbol{s}_u, \boldsymbol{s}_v \rangle\right)}{\sum_{w \in \mathcal{N}_u \cup \{u\}} \exp\left(\langle \boldsymbol{s}_u, \boldsymbol{s}_w \rangle\right)}, \quad \alpha_v^v = \frac{\exp\left(\operatorname{tr}(\boldsymbol{V}_u^\top \boldsymbol{V}_v)\right)}{\sum_{w \in \mathcal{N}_u \cup \{u\}} \exp\left(\operatorname{tr}(\boldsymbol{V}_u^\top \boldsymbol{V}_v)\right)} \tag{24}$$

The product for scalars and vectors are standard inner product and Frobenius product, respectively.

## 5 EXPERIMENTS

To demonstrate the basic rationale in perceiving angular features of our $\vec{\mathbf{W}}$-Perceptron, we design a synthetic task (Section 5.1). We then conduct experiments on multiple benchmarks, including node-level tasks: Residue Identification (**RES**) (Section 5.2), Computational Protein Design (**CPD**) (Section 5.3), and graph-level tasks: Model Quality Assessment (**MQA**) (Section 5.4). Ablation studies are also included (Section 5.5). More details are documented in the Appendix Section B.

### 5.1 SYNTHETIC TASK

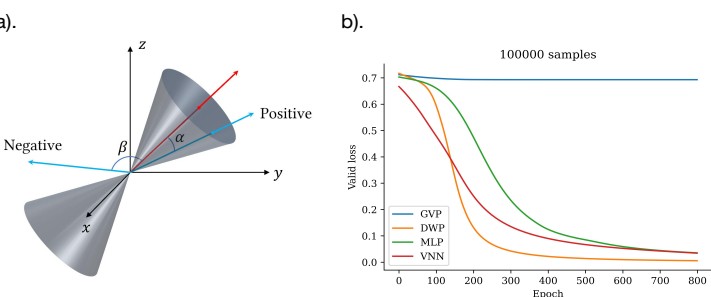

Figure 3: a). **The synthetic study**. The vector in red is the anchor used to calculate the ground truth labels. Vectors falling into the cone are positive samples, whereas outside points are negative. b). The validation Binary Cross Entropy loss of compared methods.

**Datasets.** We test the performance of different perceptron designs on a synthetic binary classification dataset with angular. We sample a random unit vector $v_m \in \mathbf{R}^3$ as the anchor vector. Then each vector $v_i$ in the space is labeled positive if its angle between the anchor vector $a_{mi} \leq \pi/4$, and negative otherwise. We generate $100,000$ random vectors with $50\%$ positive samples and $50\%$ negative samples. For a fair comparison, we restrict the number of parameters and hyperparameters of different models to approximately the same.

**Results.** As shown in Figure 3, in general, our $\vec{\mathbf{W}}$-Perceptron (**DWP**) converges faster than other models and achieves the lowest validation loss. We also notice that the Geometric Vector Perceptron (**GVP**) cannot converge given a sufficient number of epochs due to the poor capability of perceiving angulars by only using linear combinations of vector features. In comparison to GVP, Vector Neural Network (**VNN**) performs better based on its non-linear ReLU operation for vector features. Simple Multi-Layer Perceptron (**MLP**) can also capture useful information according to its universal approximation ability. The superior performance demonstrates that our $\vec{\mathbf{W}}$-Perceptron has a better ability on perceiving potential geometric features in space.

### 5.2 RESIDUE IDENTIFICATION

**Datasets.** Residue Identification (RES) aims to predict the identity of particular masked amino acid based on its local surrounding structure (Torng & Altman, 2017). We download $1000,000$ $(10^6)$ substructures from the ATOM3D project (Townshend et al., 2021), which are originally derived from 574 proteins from the PDB (Berman et al., 2000). The entire dataset is split into training, validation, and testing datasets with the ratio of $80\%$, $10\%$, and $10\%$. There are no two proteins with similar structures in the test and non-test datasets based on the CATH 4.2 topology class at the domain level (Orengo et al., 1997).

**Metrics.** We use classification accuracy to evaluate model performance.

**Baselines.** The **3D-CNN** encodes the positions of the atoms in a voxelized 3D volume. We also compare GNN variants such as **GCN** , **GIN** and **GAT** with our **OA-GCN**, **OA-GIN** and **OA-GAT** models. We also include **GVP-GNN** as well.

**Results** As shown in Table 1, **GCN**, **GIN** and **GAT** almost cannot accurately identify the type of amino acids, suggesting the lack of ability of conventional GNN models to capture 3D geometric information. **3D-CNN** performs better due to the power of the voxelization technique. Our models

outperform other models, suggesting OA-GNNs could help represent protein substructure geometries better.

Table 1: Results of different RES methods on ATOM3D.

| Model | 3D-CNN | GCN | GIN | GAT | GVP-GNN | OA-GCN | OA-GIN | OA-GAT |
|---|---|---|---|---|---|---|---|---|
| Acc % | 45.1 | 8.2 | 9.1 | 12.4 | 48.2 | 50.2 | 50.8 | 49.2 |

Table 2: Ablation studies.

| Model | No DW | No Int | No Equi |
|---|---|---|---|
| Acc % | 47.3 | 47.7 | 33.0 |

## 5.3 COMPUTATIONAL PROTEIN DESIGN

**Datasets.** Computational Protein Design (CPD) predicts the native protein sequence of a given backbone structure. Specifically, we focus on two databases: CATH 4.2 (Ingraham et al., 2019) organizes proteins in a hierarchical structure (Orengo et al., 1997). Following prior works, there are $18,024$ protein chains in the training set, $608$ in the validation set, and $1,120$ in the test. TS50 dataset (Li et al., 2014) is a relatively old benchmark consisting of only $50$ protein structures widely used in biology communities. [1].

**Metrics.** Native Sequence Recovery Rate is adopted to evaluate the prediction performance, which compares the predicted sequence with the ground truth sequence at each position and calculates the proportion of the correctly recovered amino acids (Li et al., 2014). The perplexity score (Jelinek et al., 1977) evaluates whether a model could assign a high likelihood to the test sequences (Ingraham et al., 2019).

Table 3: Results of different CPD methods on the CATH 4.2.

| Model | Perplexity ↓ | | | Recovery % ↑ | | |
|---|---|---|---|---|---|---|
| | Short | Single | All | Short | Single | All |
| St-Transformer | 8.54 | 9.03 | 6.85 | 28.3 | 27.6 | 36.4 |
| St-GCN | 8.31 | 8.88 | 6.55 | 28.4 | 28.1 | 37.3 |
| St-GIN | 8.03 | 8.52 | 6.15 | 27.7 | 28.4 | 38.1 |
| St-GAT | 10.86 | 10.67 | 9.89 | 26.2 | 26.8 | 35.2 |
| GVP-GNN | 7.10 | 7.44 | 5.29 | 32.1 | 32.0 | 40.2 |
| OA-GCN | **5.42** | 5.69 | 3.94 | **39.6** | 38.5 | 47.5 |
| OA-GIN | 5.84 | **5.39** | **3.85** | 38.8 | **40.1** | **47.8** |
| OA-GAT | 5.92 | 5.53 | 4.13 | 37.5 | 39.3 | 46.7 |
| No DW | 6.52 | 6.79 | 5.28 | 36.7 | 35.0 | 42.9 |
| No Int | 6.45 | 6.36 | 4.87 | 37.2 | 36.3 | 44.9 |
| No Equi | 6.21 | 6.04 | 4.64 | 36.9 | 37.1 | 45.8 |

Table 4: Results of MQA models on CASP 11 stage 2.

| Model | Average ↑ | | | Global ↑ | | |
|---|---|---|---|---|---|---|
| | $r$ | $\rho$ | $\tau$ | $r$ | $\rho$ | $\tau$ |
| VoroMQA | 0.42 | 0.41 | 0.29 | 0.65 | 0.69 | 0.51 |
| RWplus | 0.17 | 0.19 | 0.13 | 0.06 | 0.03 | 0.01 |
| SBROD | 0.43 | 0.41 | 0.29 | 0.55 | 0.57 | 0.39 |
| Proq3D | 0.44 | **0.43** | 0.30 | 0.77 | **0.80** | 0.59 |
| 3DCNN | 0.49 | 0.39 | 0.27 | 0.64 | 0.67 | 0.48 |
| Ornate | 0.39 | 0.37 | 0.26 | 0.63 | 0.67 | 0.48 |
| DimeNet | 0.30 | 0.35 | 0.28 | 0.61 | 0.62 | 0.43 |
| GraphQA | 0.48 | 0.40 | 0.42 | 0.75 | 0.72 | 0.74 |
| GVP-GNN | 0.58 | 0.33 | 0.46 | 0.80 | 0.61 | 0.81 |
| OA-GCN | 0.62 | 0.37 | 0.51 | 0.84 | 0.65 | 0.85 |
| OA-GIN | 0.63 | 0.36 | 0.52 | **0.86** | 0.67 | **0.88** |
| OA-GAT | **0.65** | 0.42 | **0.53** | 0.83 | 0.69 | 0.86 |

Table 5: Results of structure biology methods on the TS50.

| Model | Rosetta | SPIN | ProteinSolver | Wang's | SPIN2 | SBROF | ProDCoNN | St-Trans |
|---|---|---|---|---|---|---|---|---|
| Recv % | 30.0 | 30.3 | 30.8 | 33.0 | 33.6 | 39.2 | 40.7 | 42.3 |

| Model | DenseCPD | GVP-GNN | OA-GCN | OA-GAT | OA-GIN | No DW | No Int | No Equi |
|---|---|---|---|---|---|---|---|---|
| Recv % | 50.7 | 44.9 | 53.8 | **54.5** | 52.7 | 46.8 | 48.7 | 49.5 |

**Baselines.** For the CATH 4.2 dataset, we compare our models with other state-of-the-art models, including **GVP-GNN**, **Structured Transformer** and **Structured GNN** (Ingraham et al., 2019). We compare multiple machine learning approaches in the TS50 dataset, including 3D CNN-based methods (**ProDConn** (Zhang et al., 2020), **DenseCPD** (Qi & Zhang, 2020)), sequential models (**Wang's model** Wang et al. (2018), **SPIN** (Li et al., 2014), **SPIN2** (O'Connell et al., 2018)) and **GNN-based** method (Strokach et al., 2020). We also compare with classical statistic methods (Schaap et al., 2001; Cheng et al., 2019).

**Results.** As shown in Table 3, our models achieve significant improvement over other methods on the CATH dataset. This result also suggests that the the short-chain and single-chain subsets are more challenging, which is consistent with the result in (Ingraham et al., 2019). As shown in Table 5, our models also gain superior performance. These results suggest that adopting OA-GNNs is a more effective way of designing valid protein sequences.

## 5.4 MODEL QUALITY ASSESSMENT

**Datasets.** Model quality assessment (MQA) evaluates the quality of predicted protein structures (Kwon et al., 2021). The goal is to fit a model approximating the numerical metric used to compare

---

[1] As there is no canonical training set for TS50, we follow prior methods to use protein sequences with less than $30\%$ similarity with the TS50 test set from the CATH training set for training (Jing et al., 2020; Qi & Zhang, 2020)

the predicted 3D structure with the native 3D structure. For this task, We randomly split the targets in CASP5-CASP10 (Moult et al., 2014) and sample 50 decoys for each target for generating the training and validation sets and use the CASP11 stage 2 proteins as the test set to ensure no similar structures are involved. In total, there are 508 proteins for training, 56 for validation, and 85 targets with 150 decoys for the test.

**Metrics.** We regress the global GDT_TS score (Zemla, 2003), which evaluates the quality of decoys. We adopt the correlation between the predicted and real GDT_TS values to measure the performance. We use three statistical correlation metrics: Pearson's correlation $r$, Spearman's $\rho$, and Kendall's $\tau$. We calculate the correlation for each target and average all the correlations over all the targets. We also calculate the Global correlations by taking the union of all decoy sets without considering the targets (Pagès et al., 2019).

**Baselines.** **VoroMQA** (Olechnovič & Venclovas, 2017) leverages potential model with protein contact maps, while **RWplus** (Zhang & Zhang, 2010) relies on physical energy terms. **SBROD** (Karasikov et al., 2019) uses hand-crafted features. **Proq3D** (Uziela et al., 2017) employs a FCNs for regression. **3DCNN** (Derevyanko et al., 2018) and **Ornate** (Pagès et al., 2019) apply 3D CNNs for extracting meaningful features. **GraphQA** (Baldassarre et al., 2021), **DimeNet** (Gasteiger et al., 2019b) and **GVP-GNN** are the most similar models as ours which adopted GNN-based methods on protein graphs.

**Results.** As shown in table 4, our models outperform other methods on both average and global metrics except for Spearman's correlation $\rho$ where we achieve the second-best, demonstrating our models can not only evaluate the quality for the same protein but also works well across different proteins in a more general way.

## 5.5 ABLATION STUDIES

In the end, we conduct ablation experiments to study the contribution of different modules of our model on the tasks of CPD (table 3, 5) and RES (table 2). More specifically, based on OA-GCN, we replace all the directed weights with the regular linear layer with scalar weights (**No DW**), or remove the Interaction Module (**No Int**), or break the equivariance by canceling coordinate transformations in the step of message passing (**No Equi**).

In general, all three components of our model are important since removing each of them results in a significant performance drop. According to the performance gain, the directed weights are the most important module for the CPD task, as the orientational features are essential for the protein to fold to a given structure. For Residue Identity, however, equivariance message passing is the most important component, indicating the necessity of rotation equivariance in learning the representation of local protein structures.

## 6 CONCLUSION

For the first time, we generalize the scalar weights in neural networks to 3D directed vectors for better perceiving geometric features like orientations and angles, which are essential in learning representations from protein 3D structures. Based on the directed weights $\vec{\mathbf{W}}$, we provide a toolbox of operators as well as a $\vec{\mathbf{W}}$-Perceptron Unit encouraging efficient scalar-vector feature interactions. We also enforce global $SO(3)$-equivariance on a message passing paradigm using the local orientations naturally defined by the rigid property of each amino acid residue, which can be used for designing more powerful networks. Our integrated Orientation-Aware Graph Neural Networks achieve comparably better performances on multiple biological tasks in comparison with existing state-of-the-art methods.

**Limitations and future work.** While our method shows better awareness to fine geometric details, there remains a huge space for non-linearity designs upon geometric representations. In the future, we will extend our framework to other 3D learning tasks like small molecules or point clouds by learning local rotation transformations and using directed vector weights.

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

## A  Proof of Rotation Equivariance

A function $\boldsymbol{f}$ taking a 3D vector $\boldsymbol{x} \in \mathbb{R}^3$ as input is rotation equivariance, if applying ant rotation matrix $\boldsymbol{R} \in \mathbb{R}^{3\times3}$ on $\boldsymbol{x}$ leads to the same transformations of the output $\boldsymbol{f}(\boldsymbol{x})$. Formally, $\boldsymbol{f}: \mathbb{R}^3 \rightarrow \mathbb{R}^3$ is rotation equivariance by fulfilling:

$$\boldsymbol{R}(\boldsymbol{f}(\boldsymbol{x})) = \boldsymbol{f}(\boldsymbol{R}\boldsymbol{x}) \tag{25}$$

For notation consistency, we consider each row of the matrix to be an individual 3D vector and use right-hand matrix multiplication here. When performing equivariant message passing on protein graph for node $i$ with local rotation matrix $\boldsymbol{O}_i$, we first transform the vector representations of its neighbors from global reference to its local reference, that is, for a particular neighbor node $j$ with vector feature $\boldsymbol{V}_j \in \mathbb{R}^{C\times3}$, apply our DWNN layers $\boldsymbol{f}$, and transform the updated features back to the global reference. We set only one neighbor node $u_j$ for $u_i$ for simplicity. The updated vector representation $\boldsymbol{V}_i$ for node $i$ is

$$\boldsymbol{V}_i = [\boldsymbol{f}(\boldsymbol{V}_j\boldsymbol{O}_i^{\mathrm{T}})]\boldsymbol{O}_i \tag{26}$$

If we apply a global rotation matrix $\boldsymbol{R}$ to all vectors in the global frame, the local rotation matrix $\boldsymbol{O}_i$ will be transformed to $\boldsymbol{O}_i^{'} = \boldsymbol{O}_i\boldsymbol{R}$, and $\boldsymbol{V}_j$ to be $\boldsymbol{V}_j^{'} = \boldsymbol{V}_j\boldsymbol{R}$, now the output $\boldsymbol{V}_i^{'}$ is

$$\boldsymbol{V}_i^{'} = [\boldsymbol{f}(\boldsymbol{V}_j^{'}\boldsymbol{O}_i^{'\mathrm{T}})]\boldsymbol{O}_i^{'} \tag{27}$$
$$= [\boldsymbol{f}(\boldsymbol{V}_j\boldsymbol{R}\boldsymbol{R}^{\mathrm{T}}\boldsymbol{O}_i^{\mathrm{T}})]\boldsymbol{O}_i\boldsymbol{R} \tag{28}$$
$$= [f(\boldsymbol{V}_j\boldsymbol{O}_i^{\mathrm{T}})]\boldsymbol{O}_i\boldsymbol{R} \tag{29}$$

And if instead, we directly apply the rotation matrix to original output $\boldsymbol{V}_i$, we got

$$\boldsymbol{V}_i^{'} = [\boldsymbol{f}(\boldsymbol{V}_j\boldsymbol{O}_i^{\mathrm{T}})]\boldsymbol{O}_i\boldsymbol{R} \tag{30}$$
$$= \boldsymbol{V}_i\boldsymbol{R} \tag{31}$$

In other words, rotating the input leads to the same transformations to output, so we can preserve rotation equivariance for vector representations in this way. Note that scalar features remain invariant when applying global rotation, so our message-passing paradigm with global and local coordinate transformations is rotation equivariance.

## B  Experiment Details

We represent a protein 3D structure as an attributed graph, with each node and edge attached with scalar and vector features that are geometric-aware. We implement our DWNN in the equivariant message passing manner, with 3 layer Directed Weight Perceptrons for all tasks.

### B.1  Protein Features

In this paper, we use an attributed graph $\mathcal{G} = (\mathcal{V}, \mathcal{E})$ to represent the protein structure, where each node corresponds to one particular amino acid in the protein with edges connecting its k-nearest neighbors. Here we set $k = 30$. The node features $\mathcal{V} = \{v_1, .., v_N\}$ and edge features $\mathcal{E} = \{e_{ij}\}_{i\neq j}$ are both multi-channel scalar-vector tuples with scalar features like distances and dihedral angles and vector features like unit vectors representing particular orientations.

A node $v_i$ represents the $i$-th residue in the protein with scalar and vector features describing its geometric and chemical properties if available. Therefore, a node in this graph may have multi-channel scalar-vector tuples $(\boldsymbol{s}_i, \boldsymbol{V}_i)$, $\boldsymbol{s}_i \in \mathbb{R}^6$ or $\mathbb{R}^{26}, \boldsymbol{V}_i \in \mathbb{R}^{3\times3}$ as its initial features.

- **Scalar Feature.** The $\{\sin, \cos\} \circ \{\psi, \omega, \phi\}$. Here $\{\psi, \omega, \phi\}$ are dihedral angles computed from its four backbone atom positions, $C\alpha_{i-1}, N_i, C\alpha_i, N_{i+1}$.
- **Scalar Feature.** A one-hot representation of residue if the identity is available.
- **Vector Feature.** The unit vectors in the directions of $C\alpha_{i+1} - C\alpha_i$ and $C\alpha_{i-1} - C\alpha_i$.
- **Vector Feature.** The unit vector in the direction of $C\beta_i - C\alpha_i$ corresponds to the side-chain directions.

The edge $e_{ij}$ connecting the $i$-th residue and the $j$-th residue also has multi-channel scalar-vector tuples as its feature $(\boldsymbol{s}_{ij}, \boldsymbol{V}_{ij}), \boldsymbol{s}_{ij} \in \mathbb{R}^{34}, \boldsymbol{V}_{ij} \in \mathbb{R}^{1 \times 3}$

- **Scalar Feature.** The encoding of the distance $\|C\alpha_j - C\alpha_i\|$ using 16 Gaussian radial basis functions with centers spaced between 0 to 20 angstroms.

- **Scalar Feature.** The positional encoding of $j - i$ corresponding the relative position in the protein sequence.

- **Scalar Feature:** The contact signal describes if the two residues contact in the space,1 if $\|C\alpha_j - C\alpha_j\| \leq 8$ and 0 otherwise.

- **Scalar Feature.** The H-bond signal describes if there may be a H-bond between the two nodes calculated by backbone distance.

- **Vector Feature.** The unit vector in the direction of $C\alpha_j - C\alpha_i$.

## B.2   MODEL DETAILS

**Synthetic Task.** We uniformly sample points on the sphere and the ratio of positive and negative samples is 1:1. We consider the length of each vector as one channel scalar feature $\in \mathbb{R}$ and the vector itself as one channel vector feature $\in \mathbb{R}^{1 \times 3}$. The VNN and GVP models in this experiment are set to 3 layers. And our DWP is only 1 layer, consisting 1 Directed Linear Module, 1 NonLinear Module, and 1 Directed Interaction Module. We also train a 3-layer MLP by concatenating the scalar and vector feature as a four-channel scalar feature $\in \mathbb{R}^4$ as input. Specifically, the parameter counts of VNN, GVP, MLP, DWP is 24,28,24 and 24. We use Binary CrossEntropy Loss for this two-class classification task. For MLP, the scalar and vector features are concatenated along the channel, which means the feature of each point is a vector in $\mathbb{R}^4$. For GVP and VNN, the input scalar and vector are represented by separate neural network components.

For all models we trained in the protein-related task, we use 128 scalar channels and 32 vector channels for each node's hidden representations, 64 scalar channels, and 16 vector channels for each edge's hidden representations. There is 4 message passing updations in implemented models, where each passing layer consists of 3 stacked DWPs. In total there are about 2.9 million parameters of our models.

**CPD.** We train our model in a BERT-style recovery target (Strokach et al., 2020), more specifically, we dynamically mask 85% residues of the whole protein and let the model recover them using structure information from its neighbors using CrossEntropy loss. During testing, we mask all the residues and recover the whole protein sequence in one forward pass, we also try to iteratively predict one residue per forwarding pass.

**MQA.** Because MQA is a regression task, we train our model using MSE loss. When testing, we compute the different types of correlations between predicted scores and ground truth.

**RES.** Residue identity requires the model to classify the center residue from a local substructure of the protein. We construct a subgraph of each local substructure according to Section B.1. To guarantee equitable comparison, we retrain all the methods based on our protein graph from scratch.

## B.3   TRAINING DETAILS

For the synthetic task, we train each model for 1000 epochs with a learning rate of $1e - 3$ and plot the training loss for evaluation.

We train our models with learning rate $3e - 4$ for CPD and $2e - 4$ for MQA, a dropout rate of $10\%$ (Srivastava et al., 2014), and Layernorm paradigm. We train all the models on NVIDIA Tesla V100 for 100 epochs for each task.

Our model uses 4-layer message passing, where each message passing consists of 3-layer DWP. The total model consists of about 2.9 million parameters.

## B.4 BASELINE DETAILS

We obtain most of our baseline numerical results from other benchmark papers. But because some models haven't been evaluated on some of the datasets we used, we reimplement them following the same model complexity and training hyperparameters as our models.

In **the synthetic dataset**, we download the source code from (Deng et al., 2021) and (Jing et al., 2021), both of which use CrossEntropy loss. The number of parameters of VNN, GVP, MLP, and DWP is $24, 28, 24$, and $24$, respectively.

For the RES task **Table 1**, we implement GCN, GIN, and GAT models, which are trained and evaluated with the same settings as our models. Following the source code from (Townshend et al., 2021; Jing et al., 2021), we also retrain 3D CNN and GVP-GNN. To make a fair comparison, we keep the model complexity (e.g. hidden dimensions, layer numbers) and hyperparameters (epochs, random seed) of these baselines the same as our models.

For the CPD task in **Table 3**, we report the result of St-Transformer from (Ingraham et al., 2019), St-GCN, and GVP-GNN from (Jing et al., 2020). We modify the source code from Structed-Transformer(Ingraham et al., 2019) to get the result of St-GAT and St-GIN by replacing the original attention layer in the encoder and decoder module with Graph Attention Network and Graph Isomorphism Network. The hidden dimensions and layer numbers are kept the same as our models. In **Table 5**, we adopt most of the results from (Jing et al., 2020) except running the St-Transformer model on the TS50 dataset.

For MQA in **Table 4**, we merge the results of the same task from (Jing et al., 2020) and Structed-Transformer (Ingraham et al., 2019). As GVP-GNN only reports three of six of our metrics, so we reimplement GVP-GNN and DimeNet on our benchmark. Same as the above model, the hidden dimensions and layer numbers are kept the same as our models.

For our ablation models, we concatenate scalar and vector features as input and replace all the directed weights with the regular linear layer with scalar weights to get the No DW model. By removing the Interaction Module lying in the final layer of each perceptron, we obtain the No Int model. We also replace local frames with unity matrices to break the equivariance in the message passing part (No equivariance).

In the FOLD and REACT tasks, Kipf et al. and Diehl et al. construct the input graph using the protein's contact map and use pair-wise distances of residues as edge features. Baldassarre et al. use the spatial features including the dihedral angles, surface accessibility, and secondary structure type of each residue's node feature. They also use distances, sequence distances, and bond types for edge features. Hermosilla et al. consider more fine-grained atom-level spatial information, e.g. covalent radius, van der Waals radius, and atom mass, which leads to better performance. Gligorijevi et al. refer to different types of contact maps and geometric distances for the input of GNN. All GNN and CNN baselines make use of 3D information in proteins.

## C ADDITIONAL RESULTS

### C.1 MULTIPLE RUNS ON RES TASK

We conduct different runs on RES dataset for both baseline and our models. As shown in Table 6, we report the mean and std results of each method across five different random seeds, our models are still doing best among other methods and have stable, robust performance with few deviations.

Table 6: Results across five different random seeds of models on RES dataset (Mean ± Std).

| Model | 3D CNN | GCN | GIN | GAT | GVP-GNN | OA-GCN | OA-GIN | OA-GAT |
|-------|--------|-----|-----|-----|---------|--------|--------|--------|
| Acc % | 45.0±0.4 | 8.2±0.3 | 9.2±0.5 | 12.4±0.2 | 48.4±0.4 | 50.2±0.2 | 50.8±0.1 | 49.2±0.3 |

### C.2 ITERATIVE PREDICTION ON CPD TASK

We tested the sequential decoding of the prediction from the N-side to the C-side and reported the native sequence recovery as shown in Table 7,

Table 7: Iterative prediction results of our models on both CATH (Short, Single, All) and TS50 dataset.

| Model | Short | Single | All | TS50 |
|---|---|---|---|---|
| OA-GCN (iter) | 34.1 | 35.7 | 45.3 | 50.9 |
| OA-GIN (iter) | 34.4 | 37.2 | 45.7 | 52.2 |
| OA-GAT (iter) | 36.0 | 36.5 | 36.5 | 49.6 |

The results have shown that the sequential decoding approach basically caused a degradation of the results, but because we introduced the directed weight and geometric operations, our results are still better than other benchmarks. We have added these results in the supplementary material.

## C.3 PROTEIN FUNCTION CLASSIFICATION

**Datasets.** Fold Classification (FOLD) predicts the structure category of a given protein structure. We choose the SCOP v1.75 dataset collected by Murzin et al. (1995) that organizes structures into 3-layer hierarchical classes. In total, there are $16,712$ proteins covering 7 major structural types with $1,195$ identified folds. We adopt a reduced dataset from Hou et al. (2018), and remove homologous sequences between test and training data sets at Family (**Fam**), Superfamily (**Sup**) or **Fold** levels, resulting in three different classification tasks.

The Enzyme-Catalyzed Reaction Classification (REACT) study is a very similar task as it requires to classify the EC number Webb et al. (1992) of a catalyzed enzyme based on the protein structures. Therefore, we put these two tasks side-by-side to evaluate the performance of different methods. We use the dataset collected by (Hermosilla Casajus et al., 2021), containing $37,428$ proteins from $384$ types of Enzyme-Catalyzed classes and we split them into training, validation set, test set and ensure no sequence or structure overlaps across different sets. In total, there are $29,215$ structures for training, $2,562$ for validation, and $5,651$ for test.

**Metrics.** Following standard benchmarks Diehl (2019); Hermosilla Casajus et al. (2021), we use the classification accuracy for evaluating the prediction performance.

**Baselines.** We compare with **CNN-based** models and **GNN-based** models which learn the protein annotations using 3D structures from scratch. For fair comparison, we also do not include LSTM-based or transformer-based methods, as they all pre-train their models using millions of protein sequences and only fine-tune their models on 3D structures Bepler & Berger (2018); Alley et al. (2019); Rao et al. (2019); Strodthoff et al. (2020); Elnaggar et al. (2020).

**Results.** As shown in Table 8, our models demonstrate comparable prediction performance in both tasks across different levels of the hierarchy. Classifying the fold and enzyme classes based on the protein structure is one of the key problems in structural biology and our proposed models can be used as new tools to conduct function annotations for new proteins.

For FOLD and REACT tasks, all GNN-based baselines use 3D structure information:

- Kipf & Welling (2016) and Diehl (2019) construct the input graph using the protein's contact map and use pair-wise distances of residues as edge features.
- Baldassarre et al. (2021) use the spatial features including the dihedral angles, surface accessibility, and secondary structure type of each residue's node feature. They also use distances, sequence distances, and bond types for edge features.
- Hermosilla Casajus et al. (2021) consider more fine-grained atom-level spatial information, e.g. covalent radius, van der Waals radius, and atom mass, which leads to better performance.
- Gligorijević et al. (2021) refer to different types of contact maps and geometric distances for the input of GNN.

Despite not using special training techniques, pre-training tasks, and fine-tuning of hyperparameters, our models demonstrate comparable prediction performance in both tasks across different levels of the hierarchy. Especially with the introduction of our directed weight with equivariant message passing, it is a big improvement over the normal GNN.

Table 8: Results on Fold and React classification

|  | Architecture | FOLD | | | REACT |
| --- | --- | --- | --- | --- | --- |
|  |  | Fold | Sup | Fam |  |
| Hou et al. (2018) | 1D ResNet | 17.0% | 31.0% | 77.0% | 70.9% |
| Derevyanko et al. (2018) | 3D CNN | 31.6% | 45.4% | 92.5% | 78.8% |
| Kipf & Welling (2016) | GCN | 16.8% | 21.3% | 82.8% | 67.3% |
| Diehl (2019) | GCN | 12.9% | 16.3% | 72.5% | 57.9% |
| Hermosilla Casajus et al. (2021) | GCN | 45.0% | 69.7% | 98.9% | 87.2% |
| Baldassarre et al. (2021) | GCN | 23.7% | 32.5% | 84.4% | 60.8% |
| Gligorijević et al. (2021) | LSTM+GCN | 15.3% | 20.6% | 73.2% | 63.3% |
| Our method | OA-GCN | 31.2% | 39.8% | 84.8% | 76.0% |
| Our method | OA-GIN | 32.8% | 38.3% | 85.2% | 76.7% |
| Our method | OA-GAT | 29.9% | 36.6% | 79.0% | 77.2% |

# D    COMPARISONS TO PRIOR WORKS

## D.1    COMPARISON TO ANGULAR-EXPLICIT MODELS

There are several works that explicitly model angles/torsion angles in GNNs. DimeNet(Gasteiger et al., 2019b) uses physical-informed RBF and SBF functions to represent distances and angular information. SphereNet Liu et al. (2021) further employs a 3D meaningful spherical coordinate system to represent molecular torsional angles. However, these methods rely on invariant scalar input features for remaining equivariance, and during convolution, these scalar features (distances, angles) are just used for updating hidden representations and stay the same in the next layer updating.

On the technical level, our model also has several differences from other methods when compared with existing studies to consider backbone torsion angles (DimeNet, SpehreNet, etc.), as follows:

1. RBF-embedded distances and Trigonometric-embedded angles are adopted in the input feature engineering part (see supplementary material). To better represent the 3D geometries, we also introduce several normal vectors as input node features and edge characteristics. The scalar representations are invariant and vectors are equivariant. But these methods don't consider vector representations and only focus on invariant scalar features.

2. We capture fine-grained geometries with the help of vector representations and directed weight perceptrons. And representations are updated layer-by-layer. Our directed weights are learnable to be expandable for complex reasoning capabilities. But these methods only embed angles and distances into edge features during message updating and lack geometry meaningful operations. They also don't have directed weights, which are key contributions to our work.

3. We maintain equivariance with the help of local coordinate transformations on vector representations equipped with rigid residue positions. But these methods don't consider equivariant vectors, they only condition on invariant scalars. SphereNet does use local spherical systems, but they are used for calculating relative locations of atoms, seen as a feature engineering part.

4. We aim to design a new type of vector neurons for geometric learning and a general message passing paradigm for 3D graphs. We have shown our versatility in any existing graph learning frameworks. However, these methods are designed specifically for molecular graphs and are hard to extend for other 3D point sets.

## D.2    COMPARISON TO OTHER EGNN WORKS

Local-global frame transformation is an important technique for 3D object learning (Petrelli & Di Stefano, 2011; Yang et al., 2018; Luo et al., 2022) and is inevitable in the discussions of 3D

equivariance. Our proposed framework focuses on its computation in the latent space together with our vector-based feature representations, while many existing works only apply rotations to the primal space of explicit geometric/physical properties (e.g. simple cartesian coordinates). It also provides better flexibility over message-passing operators by eliminating the requirements for specific feature-update arithmetics – this also serves as an adaptor for plugging our designed feature representations into many existing GNN models while guaranteeing equivariance.

Among the EGNNs, we could not agree that our model is a simplified model from others. Specifically, some of these EGNNs could be considered a special case of our framework. For instance, E(n) GNN (Satorras et al., 2021) combines node coordinates weighted by hidden representations, which could be recognized as linearly updating of simple vector features (coordinate) and unit matrix transformations (in analogy to our back and forth transformations).

Additionally, to the best of our knowledge, there is no previous work that integrates vector-based features and directed weighted neural networks into local-global transformation frameworks. We apply transformations on hidden scalar-vector features back and forth for better representations under equivariance, which turns out to show decent performances in the protein learning tasks.

### D.3 COMPARISON TO OTHER BACKBONE-ORIENTATION MODELS

Using $sin, cos \ldots \phi, \psi, \omega$ as features are popular techniques in protein learning tasks (Ingraham et al., 2019; Jing et al., 2021; Ganea et al., 2021), but our models can have a more powerful ability to make use of these features. The meaning of Orientation-Aware GNNs is not to use torsional angles as the node and edge features, but for reasoning on them. Our real contribution is to design a new type of neural network, which has good capability in sensing these geometric features (e.g. angles, orientations) as well as capturing complex interactions. The synthetic experiment gives an intuitive example of our claims, where our designed directed weight perceptrons have more powerful in sensing and handling angular-related characteristics. Here we share an even simpler example: Assume the input contains two 3D vectors $v_1$ and $v_2$ described by their coordinates $[x_1, y_1, z_1, x_2, y_2, z_2]$, the problem we solved is that how should we let the model know the angle between $v_1$ and $v_2$. The most straightforward intuition is that if the weights are directed vectors, then we could count on the inner products between weights and input vectors to capture the angle information. Furthermore, if one wants to retrieve more complex geometric objects, e.g. normal vectors, dihedral angles, or parallel surfaces based on input coordinates, our employed cross-product operation can easily solve that. As we use scalar-vector tuple hidden representations for nodes and edges, and directed weights with geometric-meaningful operators are learnable, our models could perform more complex geometric reasoning ability and modeling capability beyond angles, especially approximating nonlinear functions on vector representations.

The most straightforward intuition is that if the weights are directed vectors, then we could count on the inner products between weights and input vectors to capture the angle information. Furthermore, if one wants to retrieve more complex geometric objects, e.g. normal vectors, dihedral angles, or parallel surfaces based on input coordinates, our employed cross-product operation can easily solve that. As we use scalar-vector tuple hidden representations for nodes and edges, and directed weights with geometric-meaningful operators are learnable, our models could perform more complex geometric reasoning ability and modeling capability beyond angles, especially approximating nonlinear functions on vector representations.

The motivation and function of residue orientations are also different from prior works. Instead of using the angles (orientations) as input features, residue backbone orientations define the coordinate frame of the whole system and are used in the equivariant message passing part for coordinate transformation. Rigid orientations play the role of bridging local and global coordinates for equivariance in our message passing part.

# E EXPLANATIONS TO SOME MODULES

## E.1 ABOUT THE $s'_{dot}$

For the $s'_{dot}$ features, it can be interpreted as the summation of the channel-wise dot product between the input features (channels of vectors) and the directed weights (a matrix of vectors, so a 3-dimensional tensor). Let us consider the $i$-th number in the output feature, it can be formulated as:

$$s'_i = \sum_{jk} W^{ijk} v^{jk} = \sum_j (w^{ij})^T v^j$$

Notice that both are 3D vectors, **the dot product can essentially reflect the cosine value between them after normalization**. Essentially, our method utilizes the dot product operation between the directed weights and the input coordinate features, and more importantly, the weight vectors are learnable. We also recommend you pay attention to other operations we introduced for reasoning in complex geometric relations and encouraging efficient scalar-vector feature interactions. For instance, the cross-product operation can create new orthogonal orientations; the interaction modules can facilitate feature blending.

## E.2 ABOUT THE SCALR-LIFTING PART

The initial motivation of scalar lifting is to enable scalar-vector interactions, which are not well addressed in previous works (for instance, VNN (Deng et al., 2021) and GVP (Jing et al., 2020) can only perform vector-to-scalar mappings with a very limited choice of operations). Unprojecting from lower dimensions to higher dimensions is indeed non-bijective on the test data themselves, but it is a widely adopted approach like the diluted convolution or the transposed convolution that succeeds in the projection onto higher dimensional features. Mapping from high dimensions to lower space and then projecting back is also a popular technique in FCNs (Kenton & Toutanova, 2019; Jumper et al., 2021), Generative Models (Wang et al., 2014), and Manifold Learning, which can be regarded as feature compression and information bottleneck. Therefore, though not unique, the learnable directed weights endow the model with the expressiveness to recover useful information hopefully after training on many data. More intuitively, the scalar lifting operator generates fixed directional vectors with different lengths during inference, which can potentially provide more information during the next round of feature processing and facilitate information flow between scalar and vector features.

The scalar lifting module serves to pass scalar information to the vector representation in our model. Though we give a geometric interpretation of the intuition, our directed weight is learnable, the operation of the model is complex and its final geometric meaning is non-linear, which is difficult to define simply. The function of the mapping is learned according to the needs of the downstream task and is not limited to reconstruction or compression. From the experimental results, the network we designed does achieve excellent performance, demonstrating our strong geometric perception and representation capabilities.

