# OpenReview forum: "Orientation-Aware Graph Neural Networks for Protein Structure Representation Learning"
_ICLR.cc/2023/Conference — Submitted to ICLR 2023_

### Official Review · Reviewer_kS63 · 2022-10-21

**Confidence:** 3
**Correctness:** 4
**Technical Novelty And Significance:** 2
**Empirical Novelty And Significance:** 2
**Recommendation:** 6

**Clarity, Quality, Novelty And Reproducibility:**

- Clarity: the paper is clear, although it could be more precise in explaining the main contributions.
- Quality: the quality of the paper is good, and there are no major concerns regarding the design of the method.
- Novelty: the main idea of the paper is novel, sensibly extending previous work. However, my suggestion is to compare the proposed method with VNNs in a more in-depth way (not just on one synthetic experiment, and also ensuring that the comparison is fair).
- Reproducibility: it should be possible to implement the proposed method from the description, and experimental details are reported in the appendix.

**Strength And Weaknesses:**

**Strengths**:

- The paper is interesting and tackles an important problem. I expect that the paper could be of interest to the GNN, computer vision, and computational biology communities.
- The results are strong and there are no concerns in terms of significance (although confidence intervals are not reported). The design of the model is validated also through ablation studies.

**Weaknesses**:

- Equation (6) implies that $c \vec W \in \mathbb{R}^{C \times 3}$ for $s \in \mathbb{R}^C$ and $\vec W \in \mathbb{R}^{C \times 3}$.  How is the product $c\vec W$ defined here?
- A lot of the geometrical intuition behind the paper is never formally stated. For example:
	- Why is equivariance to rigid transformations broken in Equations 7,8?
	- What does it mean that the network can "sense the orientational features"?
	- What set of transformations is the proposed method equivariant to, that VNNs weren't before?
	- What does it mean that the proposed method is "more adaptive to the geometrically meaningful features?"
- In Experiment 5.1, do all models have a comparable number of parameters? DWP uses significantly more parameters than VNN, so this should be factored into the comparison.
- Why is there no comparison with VNN in Experiment 5.2? Is there some reason that makes it impossible to design a GNN with vector neurons? This is important because GVP is the worst-performing baseline in Experiment 5.1, so the more interesting comparison is between OA and VNN.
- Typos:
	- Missing parenthesis in the numerator of Eq. 24, right.

**Summary Of The Paper:**

This paper presents a class of SO(3)-equivariant neural networks called Orientation-Aware Graph Neural Networks (OAGNNs).
The model is based on specific layers acting on scalar and vector features, designed to ensure better expressive power (compared to previous works) while maintaining SO(3) equivariance.

Compared to previous related work, the main novelty of the proposed method is to have order-3 tensors of learnable weights acting on the features, effectively expanding classical perceptron weights from scalars to oriented vectors in 3d.

In the experimental section, the authors focus primarily on tasks related to structural biology.
This is a natural playground for SO(3) equivariant models, since many important characteristics of proteins can be described in terms of local frames irrespective of the global orientation of the molecules.

All results show that the proposed method is significantly better than the tested baselines, on almost all tasks and metrics considered.

**Summary Of The Review:**

There are no major concerns with the paper, but there is margin for improvement. Extending the comparison with previous work is particularly important, since it is not clear whether the proposed method is simply making use of the extra parameters compared to VNN.

I have recommended a weak acceptance, conditional on the authors addressing my concerns above.

---

> ### Author Response · Authors · 2022-11-18
> **Response to Reviewer kS63, Part 1**
>
> Thank you for your insightful and constructive comments as well as your appreciation of our work in **effectively expanding classical perceptron weights with strong results.** Below are some clarifications and answers to your questions. If our response does not fully address your concerns, please post additional questions and we will be happy to have further discussions.
>
> ## Q1:  Explanation of Equation 6
>
> Formally, the $ij$ th entry of $s\vec{W}$ is $s_i\vec{W}_{ij}$.
>
> We have added this new definition to our manuscript. The initial motivation of scalar lifting is to enable scalar-vector interactions, it serves to pass scalar information to the vector representation in our model.
>
>
>
> ## Q2: Why is equivariance to rigid transformations broken in Equation 7,8? What set of transformations is the proposed method equivariant to, that VNNs weren't before?
> A function $f$ taking a 3D vector $x \in R^3$ as input is rotation equivariance if applying ant rotation matrix $R \in R^{3\times 3}$ on $x$ leads to the same transformations of the output $f(x)$. Formally, $f:R^3 \to R^3$ is rotation equivariance by fulfilling: $R(f(x)) = f(Rx)$.
>
> **Both Equation 7 and Equation 8 are rotation equivariant.** For Equation 7, as scalar features are invariant to 3D transformations, they remain equivariant. And for Equation 8, the linear combinations of vectors are equivariant to rotations. **However, if one wants to apply more complex functions on 3D latent space, it'll break the equivariance**, for example, applying element-wise ReLU function on vector representations is not equivariant. We have rephrased our descriptions in the main text:
>
> `While these operators enable a more flexible network design, equivariance to rigid transformations is broken if considering more complex functions beyond linear combinations.`
>
> **Both VNNs and our models are SO(3) equivariance**, which means they are equivariant to rotation transformations in 3D space. However, we can perform beyond linear transformations on vector representations, and our equivariance message passing scheme brings us great flexibility. We have included a formal proof of equivariance of our method in Appendix part A.
>
>
>
> ## Q3: What does it mean that the network can "sense the orientational features"?
>
> It means our model can better capture and represent 3D angular-related features, e.g. dihedral angles of the protein backbone, orthogonal vectors of the point cloud, and interatomic moments. We use this designation (*orientational features*) to **distinguish common scalar features like distances or charges**. In the synthetic task, we have shown that our model can do well in directly perceiving angulars between points.
>
> More specifically, let's consider our $s_{dot}'$ operation in Equation 4, where the use of dot product can essentially reflect the cosine value between vector features. And as our directed weights are learnable to be expandable for complex reasoning capabilities.  **Other operations we designed also have specific geometric intuitions for sensing and processing orientational features.** Please refer to Section 3 and Appendix E for detailed explanations.
>
> ## Q4: What does it mean that the proposed method is "more adaptive to the geometrically meaningful features?"
>
> Sorry for the confusion.
>
> Here we mean that our model is more expressive in terms of modeling the geometric features of the 3D objects.
> **In 3D space, we can use multiple 3D vectors to describe some geometric objects.** As we introduce 3D-directed vectors for both neuron and weight matrices, our method can directly represent and process these features. Furthermore, as our latent space is also vectorized and directed weights are learnable, we can perform complex combinations of input features for downstream applications. To give an intuitive example, if we want to predict a vector C based on another two vectors A and B, and C is out of the plane constructed by the two given vectors. The outputs of only linear combinations of A and B are hard to predict C by definition. To address this problem, We boost the expressiveness of GVP by extending both neurons and weights to high-order 3D vectors. The synthetic task of classifying points based on angles is another great example of our demonstration (see Section 5.1).

---

> > ### Author Response · Authors · 2022-11-18
> > **Response to Reviewer kS63, Part 2**
> >
> > ## Q5: Do all models have a comparable number of parameters in Experiment 5.1?
> >
> > Yes, as we explained in the appendix, we kept the number of parameters of these models the same for a fair comparison.
> >
> > `The VNN and GVP models in this experiment are set to 3 layers. And our DWP is only 1 layer, consisting of 1 Directed Linear Module, 1 NonLinear Module, and 1 Directed Interaction Module. We also train a 3-layer MLP by concatenating the scalar and vector feature as a four-channel scalar feature $\in \R^4$ as input. Specifically, the parameter counts of VNN, GVP, MLP, and DWP are $24$,$28$,$24$, and $24$.`
> >
> > This shows that the directed weights we introduced **do fundamentally improve the perceptual performance of the model, not mainly due to the additional parameters brought.**
> >
> >
> >
> >
> >
> > ## Q6: Why is there no comparison with VNN in Experiment 5.2?
> >
> > GVP compares similar tasks with us in protein-related benchmarks, while VNN was originally proposed for 3D point cloud processing. VNN only considers using vector features (the coordinates points), but scalar features such as residue types and charges are important for proteins.  However, both our model and GVP take scalar-vector tuples as input, hidden representations, and output.
> >
> > We remove the connection between scalar and vector features in the Directed Linear Module, replace the vector-nonlinearity in Non-Linearity Module with VNN-ReLU, and remove the Directed Interaction Module. The additional results of VNN on RES and CPD are as follows:
> >
> > |                      | RES Acc% | CPD Short Recovery % | CPD Single Recovery % | CPD All Recovery % |
> > | :------------------: | :------: | :------------------: | :-------------------: | :----------------: |
> > |  VNN (0.6 million)   |   41.3   |         28.6         |         25.7          |        35.2        |
> > |  VNN (3.1 million)   |   45.2   |         30.1         |         27.4          |        37.0        |
> > | OA-GCN (2.9 million) |   50.2   |         39.6         |         38.5          |        47.5        |
> >
> >
> >
> > ## Q7: Equation 24 is missing parameters on the right!
> >
> >   Thanking for pointing that out! We have fixed this typo in the revised version.

---

### Official Review · Reviewer_6r4X · 2022-10-24

**Confidence:** 2
**Correctness:** 4
**Technical Novelty And Significance:** 4
**Empirical Novelty And Significance:** 4
**Recommendation:** 8

**Clarity, Quality, Novelty And Reproducibility:**

I found the presentation clear and the results look convincing.   I am not really able to judge its originality.

**Strength And Weaknesses:**

Strengths: simple ideas, broad experimental testing, good performance improvement.

Weaknesses: very empirical, leaves much for future work.

**Summary Of The Paper:**

This is a work on rotationally equivariant feed forward networks for applications in protein folding and related tasks.
A number of equivariant operations which appear useful for these applications are introduced, such as a cross product operator.
Experiments are done on several useful tasks using protein structure databases, and significantly improved results are reported.

**Summary Of The Review:**

Valuable improvements to ML for protein folding.

---

> ### Author Response · Authors · 2022-11-18
> **Respons to Reviewer 6r4X**
>
> Thank you so much for your recognition and enjoyment of **our clear presentation and convincing results**.
>
> If you have any questions, please feel free to communicate and discuss them with us further :) We are happy to solve that.

---

### Official Review · Reviewer_EZjF · 2022-11-05

**Confidence:** 4
**Correctness:** 3
**Technical Novelty And Significance:** 3
**Empirical Novelty And Significance:** 2
**Recommendation:** 5

**Clarity, Quality, Novelty And Reproducibility:**

The proposed method is novel and the results look reasonable, but the paper lacks clarity and is missing some important details and justification of some of the modeling decisions.

**Strength And Weaknesses:**

This paper introduces a novel concept, but the presentation, design choices, and testing methodology could all be much clearer. Moving some of the information from the appendix to the text would clarify the role of various mechanisms presented. For example, discussing more directly the scalar and vector feature constructions earlier in the text and not leaving them to the appendix would justify the use of the separate scalar features in the first place, which may otherwise seem extraneous when compared to feature vectors that could be magnitudes larger. Second, it isn't clear why the Directed Interaction model is necessary. Its stated purpose is to establish a connection between the scalar and vector features, which seems redundant given the operations blending scalar and vector features in the Directed Linear module. Clarification on what this module provides that the latter does not would be helpful.

The experimental results generally seem reliable and convincing, but it is not immediately clear what configurations of models are being compared to each other. It even seems in some places that conflicting information is given i.e. in B.2 the DWP model is 1 layer for the Synthetic Task but B.3 "Our model uses 4-layer message passing, where each message passing consists of 3-layer DWP". The latter further seems redundant with "there is 4 message passing...3-layer networks." One can tease these statements/cases apart, but more concise wording and clearer statement of model architecture/hyperparameters would be beneficial. Further, the synthetic task does not seem convincing without more information about how the points on a sphere were sampled, and statistics such as the variances of the positive and negative samples. Finally, it seems rather astounding that an MLP can match the performance of the most sophisticated models with a comparable number of weights on this task. The universal approximation theorem is relevant for arbitrarily deep and/or wide networks, so this is not sufficient justification. Similarly, it appears that GVP fails completely on this synthetic task, but is tested in later tasks whereas VNN performs well and is excluded without explanation.

The visualizations are informative but the table layout (tables 3-5) is confusing. There are also many citations to the arxiv version of papers that have been published in ML conference proceedings or other venues. The references should be carefully checked and corrected.

Some specific questions:
1. Why was GVP chosen over VNN for testing on benchmarks?
2. What applications outside of protein embedding is this work relevant to?
3. Why is the Directed Interaction module necessary when feature blending already occurs in the Directed Linear modules?
4. How are the k-nearest neighbors computed? Is the full N x N distance matrix available?
5. How could MLPs produce similar results as more efficient graph models with the same number of parameters?
6. Is there rigorous rationale for averaging Pearson r, Spearman rho, and Kendall tau? Was there consideration of weighting the factors? Is there any redundancy between the metrics?
7. How exactly were the synthetic task vectors sampled? Were the positive and negative areas of equal area? was the negative area less densely sampled?
8. Will code and trained models be released?

**Summary Of The Paper:**

The authors propose a graph neural network architecture that uses vectorized neurons, extensive hidden feature mixing operations, and global/local reference frame transformations to produce protein embeddings that are sensitive to or "aware of" residue-specific orientations within the larger macromolecule. They first introduce Directed Weights, combinations of multidimensional tensors that generalize scalars to vectors and vectors to higher-dimensional tensors. They then describe the Directed Weight Perceptron module, which performs numerous scalar and vector operations such as linear multiplication, dot products, and cross products. Armed with networks that more naturally describe data and relationships in 3D space, the authors next describe their SO(3) equivariant message passing paradigm. They then describe a few variants of their model adapted to mimic the architectures of recent graph models such as GCNs, GINs, and GATs. Finally, the authors demonstrate their models' performance on a number of protein-related tasks.

**Summary Of The Review:**

An interesting method, but the presentation lacks clarity and several design decisions are not well justified.

What would improve my score: add clarifications and additional details.

---

> ### Author Response · Authors · 2022-11-18
> **Response to Reviewer EZjF, Part 1**
>
> We would like to express our sincerest gratitude for your appreciation of our **introduced novel concepts** and for your comments on our work. We have now **extensively revised and expanded the manuscript** in response to all of the insightful suggestions.
>
> ## Q1: Discuss more directly the scalar and vector feature constructions in the main text.
>
> In the original submitted manuscript, we have listed all the scalar and vector features in the section **Appendix B.1**. In Section 3, we would like to design a generally new type of perceptron, so we don’t include specific constructions at the beginning of this section. We have added to demonstrate that the scalar and vector features are separate, and the channel numbers of scalars are larger than the vectors so that they are compatible, as follows,
>
> `For simplicity, we set the channel numbers for scalar and vector features as the same, but they can be different. In practice, scalar, vector features are constructed based on a given protein structure and are kept separate. To make them compatible, the channel numbers of scalars are larger than the vectors`
>
> Please let us know if there are any other issues related to feature construction beyond the contents in **Appendix B.1**.
>
>
>
> ## Q2: It isn't clear why the Directed Interaction model is necessary. Why is the Directed Interaction module necessary when feature blending already occurs in the Directed Linear modules?
>
> Generally, we design Directed Linear and Directed Interaction Module for communicating between scalar and vector representations, but **they perform that in different ways.** Directed Linear modules perform linear transformations by establishing four updating ways (from scalar to scalar, scalar to vector, vector to scalar, and vector to vector). These geometric meaningful operations help to perceive and process 3D information.
>
> However, Directed Interaction modules are designed in another way. As shown in Equations 12 and 13, here we **construct residual connection from vector to a scalar, and weight vector features with a scalar.** As discussed in our main text,
>
> `Specifically, Equation 12 dynamically determines how much the output should rely on scalar and vector representations, and Equation 13 weights a list of vectors using the scalar features as attention scores.`
>
> To give a more intuitive example, in protein, if an amino acid is nonpolar and electric neutral, then its side-chain angles and atom coordinates are not important to determine the biological function of this protein. Since 3D angle features and 3D coordinate features introduce more features and potentially more noise, it is necessary to use scalar features to directly weigh the importance of each 3D feature before passing it to the next layer.
>
> In addition, we discussed if **this Interaction Module is necessary for our ablation studies**, basically, it can boost performance, but it also depends on specific tasks. Generally, the three modules we designed **can be differently combined in any order and anyway.** We aim to provide a toolbox of directed weights for application. The choice which we consider in DWP is just one of that.
>
>
>
> ## Q3: Clarifications about the experiment settings of our models.
>
> We use different settings on the synthetic task and other protein benchmarks.
>
> In the synthetic task, we just want to test the performance of our directed weight perceptrons when processing angle-related tasks. There is no GNN but only perceptrons in this task.  **The 1-layer DWP consists of 1 Directed Linear Module, 1 NonLinear Module, and 1 Directed Interaction Module.**
>
> In our benchmarks, we consider an integrated **message-passing module constructed by 3 stacked Directed Weight Perceptrons.** Then the whole model in the experiment is constructed by **4 stacked message-passing modules.** In other words, we perform 4 message passings when updating node and edge hidden representations, and each message passing is performed by 3 stacked DWP. Please see Eq. 18,19 for more formal explanations. We have rephrased our demonstrations.  We have rephrased our demonstrations in Appendix.
>
> ## Q4: In the synthetic task, how the points on a sphere were sampled, and statistics such as the variances of the positive and negative samples.
>
> We **uniformly sample points on the unit sphere based on normal distribution** and the ratio of positive and negative samples is 1:1. We have added this description in the Appendix.

---

> > ### Author Response · Authors · 2022-11-18
> > **Response to Reviewer EZjF, Part 2**
> >
> > ## Q5. How could MLPs produce similar results as more efficient graph models with the same number of parameters?
> >
> > The reason is that to run MLP, we need to concatenate both the scalar and vector features together as input of MLP. In this way, MLP can approximate this function as we give enough rotation and translation of 3D inputs. However, for GVP and VNN, the input scalar and vector are represented by separate neural network components. Since they cannot efficiently represent vector features, they fail to further integrate such vector feature representations with scale feature representations. Even though MLP can roughly approximate this function, there is still a clear gap in completely modeling this data for this task as DWP.
> >
> > ## Q6: The references should be carefully checked and corrected.
> >
> > We have updated our references in the revised manuscript.
> >
> >
> >
> >
> > ## Q7. Why was GVP chosen over VNN for testing on benchmarks?
> >
> > The reason why we choose GVP as compared to baselines in benchmarks is that GVP is designed for protein-related tasks. Both our model and GVP take scalar-vector tuples as input, hidden representations, and output. However, VNN was originally proposed for 3D point cloud processing. **Additionally, VNN originally only considers 3D coordinates as input and doesn't include scalar features, which are also important for protein characteristics (e.g. residue types, charges)**.  It is not trivial to fuse scalar features with vector features while still keeping equivalence of vector features under the VNN architecture.
> >
> > As the reviewer suggested, we modified our model to get a version of the VNN baseline. By removing the connection between scalar and vector features in the Directed Linear Module, **replacing the vector-nonlinearity in Non-Linearity Module with VNN-ReLU [8]**, and removing the Directed Interaction Module. we have tested VNN performance on RES and CPD CATH tasks, the results are as follows:
> >
> > |                      | RES Acc% | CPD Short Recovery % | CPD Single Recovery % | CPD All Recovery % |
> > | :------------------: | :------: | :------------------: | :-------------------: | :----------------: |
> > |  VNN (0.6 million)   |   41.3   |         28.6         |         25.7          |        35.2        |
> > |  VNN (3.1 million)   |   45.2   |         30.1         |         27.4          |        37.0        |
> > | OA-GCN (2.9 million) |   50.2   |         39.6         |         38.5          |        47.5        |
> >
> >
> >
> > ## Q8. What applications outside of protein embedding is this work relevant to?
> >
> > Our model could be easily applied to other 3D molecules such as RNA 3D structures, DNA 3D structures, and Protein- Nucleic acid complexes [1].
> >
> > Furthermore, we can even make use of general frames [2] [3], or even learn frames of each point end-to-end, which means our model can be applied to other geometric 3D tasks like 3D point clouds [4] and small molecules [5].
> >
> >
> >
> > ## Q9. How are the k-nearest neighbors computed?
> >
> > The full N x N distance matrix is available, as the input protein structure includes the coordinates of each atom. The k-nearest neighbors are computed based on the Euclidean distances in the 3D space. We have added this detail in the revised manuscript.
> >
> >
> >
> > ## Q10. Is there a rigorous rationale for averaging Pearson r, Spearman rho, and Kendall tau? Was there consideration of weighing the factors? Is there any redundancy between the metrics?
> >
> > Here we follow the evaluation metrics of prior works on protein MQA tasks [6] [7]. For each target protein, there are several predicted decoy structures labeled by their quality scores. The **Average correlation** first calculates the correlations of each target with its corresponding decoys, and then we average the correlations of each target to get the final result. And as the dataset in the MQA task is collected in an unbiased way, there is no weighted factor for each instance. We also represent the **Global correlation**, where we directly calculate correlations along all the decoys.
> >
> > **Different correlation metrics can reflect different aspects of results.** The Pearson product difference correlation coefficient is used to measure the linear correlation between two variables X and Y. The Spearman rank correlation coefficient allows for linear correlation analysis using the rank order magnitude of the two variables, which does not require the distribution of the original variables. The Kendall rank correlation coefficient calculates the order relationship between ordered variables based on the idea of synergy and is not sensitive to outliers. It is a more tolerable measure of the association between variables. There may be some redundancy between Spearman rho and Kendall tau as they are both ranking correlations.

---

> > > ### Author Response · Authors · 2022-11-18
> > > **Response to Reviwer EZjF, Part 3**
> > >
> > > ## Q11: Will code and trained models be released?
> > >
> > > We’ll release the code and dataset after the paper is accepted.
> > >
> > >
> > >
> > > ## References
> > >
> > > [1] Baek, Minkyung, et al. Accurate prediction of nucleic acid and protein-nucleic acid complexes using RoseTTAFoldNA. *bioRxiv* (2022).
> > >
> > > [2] Petrelli A, Di Stefano L. On the repeatability of the local reference frame for partial shape matching. ICCV 2011
> > >
> > > [3] Yang J, Xiao Y, Cao Z. Toward the repeatability and robustness of the local reference frame for 3D shape matching: An evaluation. IEEE Transactions on Image Processing, 2018
> > >
> > > [4] Luo S, Li J, Guan J, et al. Equivariant Point Cloud Analysis via Learning Orientations for Message Passing. CVPR 2022
> > >
> > > [5] Shi C, Luo S, Xu M, et al. Learning gradient fields for molecular conformation generation. ICML 2021
> > >
> > > [6] Olechnovič K, Venclovas Č. VoroMQA: Assessment of protein structure quality using interatomic contact areas. Proteins: Structure, Function, and Bioinformatics, 2017
> > >
> > > [7] Baldassarre F, Menéndez Hurtado D, Elofsson A, et al. GraphQA: protein model quality assessment using graph convolutional networks[J]. Bioinformatics, 2021
> > >
> > > [8] Deng C, Litany O, Duan Y, et al. Vector neurons: A general framework for so (3)-equivariant networks. CVPR, 2021

---

### Official Review · Reviewer_XTJW · 2022-11-05

**Confidence:** 3
**Correctness:** 3
**Technical Novelty And Significance:** 2
**Empirical Novelty And Significance:** 2
**Recommendation:** 3

**Clarity, Quality, Novelty And Reproducibility:**

The paper tries to enhance the representation of protein data, which is a vital problem to promote AI research on protein-related tasks. It remains unclear which role the DWP plays since it only expands the dimension of the learning parameter and no further analysis of the DWP or the feature it extracts. Compared with other related methods, we think the paper has limited contribution either to the proposed GNN model or the explainability of the directed weight matrix. The extended experiments show the model a promising work, more research should be taken to refine the mechanism that DWP makes the model aware of the orientation.


**Strength And Weaknesses:**

**Strength:**
1. The DWP is novel to combine the SO(3)-rotation and the linear transformation, unlike the VNN model, the authors also consider extracting orientation information with the weight matrix. Adding an extra dimension on the weight matrix is quite similar to the 3D convolution filters that extract 3D information from the 3D image data.
2. The synthetic experiments are a bonus to show the effectiveness of DWP and beat other related methods.
3. The proposed DWP module is compatible with other GNN models, which is quite interesting to investigate in many applications that require more orientation information.

**Weaknesses:**
1. The number of learning parameters will triple the size of the linear transform matrix and the complexity might be too big.
2. The ablation experiments should also consider keeping the model parameter almost the same as your proposed model since the DWP brings more parameters to the model than the other methods.
3. More illustrations or experiments to directly research the directed weight matrix compared with the undirected one.
4. More experiments are needed to verify the quality of the model.

**Question:**
1. The proposed method (even the illustrative figure) is similar to GVP. What is the merit of using DWP compared to GVP?
2. Is there any constraint act on the DWP to preserve some property? If we regard the directed weight matrix as the same as the normal weight matrix, we are not sure whether the orientation information in the data is extracted by the DWP module.



**Summary Of The Paper:**

This paper designs a directed weight operation to bake the SO(3)-action into the neural network by adding an extra dimension in the weight matrix. Except for the directed weight perceptrons, the paper proposes an equivariant message passing neural network combing the DWP module. The synthetic experiment shows the effectiveness of the DWP, and the experiments on protein 3D structures achieve good performance. Overall, though the experiments are able to show performance enhancement using such a directed weight matrix, the paper has a limited illustration of the DWP module. This paper also contains some typos, need to check more carefully.

**Summary Of The Review:**

The paper proposed a DWP plus SO(3)-equivariant GNNs for protein engineering. The merit of the design compared to the existing GVP and other equivariant message passing is not significant and experiments are not sufficient for illustrating the new model's effectiveness.

---

> ### Author Response · Authors · 2022-11-18
> **Response to Reviewer XTJW, Part 1**
>
> Thank you for your insightful and constructive comments and your appreciation of our work in **proposing a DWP plus SO(3)-equivariant GNNs and showing great experiment performance**. Below are some clarifications and answers to your questions. If our response does not fully address your concerns, please post additional questions and we will be happy to have further discussions.
>
> ## Q1: What is the merit of using DWP compared to GVP?
>
> The main contribution of this work is that we propose **a new neural network architecture to better model 3D objects** with angle features. GVP/VNN solves the problem that the hidden neurons are not equivariant to rotations when the input features are 3D coordinates, where they extend each hidden neuron from one scalar to a 3D vector. In this work, we find that in many cases where performance is limited of these vector-based networks.
>
> First, GVP fails to model the case where the input features involve complex 3D information especially orientational and angle features. As stated in the manuscript, if we want to predict a vector C based on another two vectors A and B, C is out of the plane constructed by the two given vectors. **The outputs of GVP are only linear combinations of A and B, so it’s hard to predict C by definition**. To address this problem, We boost the expressiveness of GVP by extending both neurons and weights to high-order 3D vectors. The synthetic task of classifying points based on angles is another great example of our demonstration (see Section 5.1).
>
> Second, one might notice that it is not trivial to achieve SO(3) equivariance when extending both hidden neurons and model weights to vectors/tensors and moving beyond simple linear transformations. To address this problem, **we develop an equivariant message-passing scheme to help integrate our new-defined geometric operations into modern GNN frameworks**. Unlike the fixed-mode GVP, our modules are plug-and-play and can even be further combined and improved.
>
> In conclusion, although numerous public works adopt GNN in 3D applications, few works are focusing on **designing new fundamental architectures and schemes**. We have discussed the differences between our methods and prior works in geometric learning, like angular-explicit models, EGNN works and backbone-orientation models, **please refer to Appendix D for detailed explanations.**
>
>
>
> ## Q2: Is there any constraint act on the DWP to preserve some property? we are not sure whether the DWP module extracts the orientation information in the data.
>
> As with other neural networks, **we do not include any prior constraints in our model.**
>
> Our understanding of this question is to ask whether the orientation information is preserved or learned by the model, which is also related to the intuition that extending a single weight to vectors could efficiently extract the orientation information. **The initial orientation information is provided as input for our model, and our model can learn to process and integrate them for downstream tasks.** As we discussed in our main text, based on directed weights, our defined geometric meaningful operators can capture angle features, e.g. the dot product operation can sense the angles and the cross-product operation can create new directions. These operators make it easy to learn the patterns of more complex geometric objects, e.g. normal vectors, dihedral angles, or parallel surfaces based on input coordinates. Furthermore, as we use scalar-vector tuples as our latent space, and directed weights are learnable, we could perform more complex geometric reasoning and modeling capability beyond angles using gradient descent to update parameters, especially approximating nonlinear functions on vector representations. One intuitive example is supposed the side-chain angle of an amino acid has to be 10 degrees to capture particular biological functions, then one hidden neural with a 3D vector weight could easily learn this information.

---

> > ### Author Response · Authors · 2022-11-18
> > **Response to Reviewer XTJW, Part 2**
> >
> > ## Q3: The number of learning parameters will triple the size of the linear transform matrix and the complexity might be too big. The ablation experiments should also consider keeping the model parameter almost the same as your proposed model since the DWP brings more parameters to the model than the other methods.
> >
> > As mentioned in the appendix, in the synthetic experiments we ensured that the parametric quantities of the various types of neural networks tested were similar, **the number of parameters of VNN, GVP, MLP, and DWP is 24, 28, 24, and 24, respectively**, and our DWP still outperformed the other models.  This is a shred of clear evidence that the DWP design improves the expressiveness of the model instead of introducing additional parameters.
> >
> > In our other experiments, we ensured that the models used had the same structure and number of parameters (about 2.9 million), whereas the other baselines we compared did not record their parameter counts, and many of the computer vision-based convolutional models had even larger parameter sizes than ours (e.g. ResNet).
> >
> > In the ablation study, following the common practice of the machine learning community, we strictly replaced the corresponding modules to ensure the comparability of the experimental results.
> >
> > Furthermore, we also try to add more parameters to GVP, specifically, we set the node hidden dimensions from 128 channels to 256 channels for scalar, and 32 channels to 96 channels for vector, and the resulting GVP model have 3.1 million parameters.
> >
> > |                      | RES Acc% | CPD Short Recovery % | CPD Single Recovery % | CPD All Recovery % |
> > | :------------------: | :------: | :------------------: | :-------------------: | :----------------: |
> > |  GVP (0.7 million)   |   48.2   |         32.1         |         32.0          |        40.2        |
> > |  GVP (3.1 million)   |    54    |         38.3         |         36.2          |        46.0        |
> > | OA-GCN (2.9 million) |   50.2   |         39.6         |         38.5          |        47.5        |
> >
> >
> >
> > ## Q4: It remains unclear which role the DWP plays and no further analysis of the DWP.
> >
> > Here, we use a simple example to show how DWP improves the expressiveness of GVP. Suppose a protein has two vector features A and B representing two side-chains, and we want to predict another vector feature C and C are not on the plane defined by A and B. GVP/VNN layer can only linearly combine A and B so the prediction has to lie in the plane defined by A and B. Such a limitation is due to in GVP/VNN no matter how you expand the number of hidden neurons, each hidden neuron can only be a 3-dimensional vector because GVP/VNN needs to achieve the translation and rotation equivariance. **Our solution to this problem is that we expand the weights to a vector so each hidden can still be a 3-dimensional vector and the output vector does not have to lie in the plane defined by the input vectors.** In this way, DWP significantly improves the expressiveness of the neural network model by extending the weights into vectors.
> >
> > We are developing a new type of neural network for capturing capture geometric information and complex interactions. After introducing our directed weight perceptrons with equivariant message passing, we have performed both intuitive examples and strong benchmark results to show the effectiveness and flexibility of each of our modules. **Our work surely endows a new direction of directed weights and equivariance based on the local frame.** For example, one can adopt our method to RNA and DNA structures as there is also the fixed frame and can extend to learn frames end-to-end for 3D point clouds. The expressiveness, optimization methods, robustness of the vector-based network, and directed weight are also worthy of study for future explorations. [1] [2] [3].
> >
> >
> >
> > ## References
> >
> > [1] Ling S et al. VectorAdam for Rotation Equivariant Geometry Optimization
> >
> > [2] Zisling H et al. VNT-Net: Rotational Invariant Vector Neuron Transformers
> >
> > [3] Assaad, Serge, et al. VN-Transformer: Rotation-Equivariant Attention for Vector Neurons

---

### Decision · Program_Chairs · 2023-01-20

**Decision:**

Reject

**Justification For Why Not Higher Score:**

The paper is promising but it needs to more systematically and more fully investigate what the proposed directed weights are doing and how they affect training/inference time.

**Justification For Why Not Lower Score:**

N/A

**Metareview: Summary, Strengths And Weaknesses:**

The paper considers the important problem of protein representation learning. Directed Weights Perceptrons are introduced to better represent geometric relation, along with an equivariant message passing algorithm. The resulting framework makes it possible to learn the global backbone structure as well as the local fine-grained relationships between amino acids. The approach is demonstrated on multiple benchmarks.

The  AC is very familiar with the problem and carefully examined the manuscript and discussion. All reviewers and AC believe that the proposed approach is interesting and promising. The author feedback and additional experiments address many of the reviewers concerns However, more work is needed to understand more precisely what the approach is doing and how the introduction of DWs affect performance.

- In particular, as suggests by reviewer Reviewer XTJW, it would be important to characterize more precisely the directed weight matrix compared with the undirected one.
- Also it is critical to compare training/inference times of the proposed approach compared to comparison approaches to get a full picture of how the Directed Weights affect the performance.

We strongly encourage the authors to pursue this work and incorporate the above comments, in addition to the points made in their responses to the reviewers.